# Rice Seed Purity Identification Technology Using Hyperspectral Image with LASSO Logistic Regression Model

**DOI:** 10.3390/s21134384

**Published:** 2021-06-26

**Authors:** Weihua Liu, Shan Zeng, Guiju Wu, Hao Li, Feifei Chen

**Affiliations:** 1School of Electric & Electronic Engineering, Wuhan Polytechnic University, Wuhan 430023, China; whliu2020@whpu.edu.cn; 2School of Mathematics & Computer Science, Wuhan Polytechnic University, Wuhan 430023, China; lihao@whpu.edu.cn (H.L.); whqg_chenff@163.com (F.C.); 3The Key Laboratory of Earthquake Geodesy, Institute of Seismology, China Earthquake Administration, Wuhan 430023, China; wuguiju@eqhb.gov.cn

**Keywords:** hyperspectral imaging, LASSO logistic regression model, wavelength band selection, grey-scale image, seed purity identification

## Abstract

Hyperspectral technology is used to obtain spectral and spatial information of samples simultaneously and demonstrates significant potential for use in seed purity identification. However, it has certain limitations, such as high acquisition cost and massive redundant information. This study integrates the advantages of the sparse feature of the least absolute shrinkage and selection operator (LASSO) algorithm and the classification feature of the logistic regression model (LRM). We propose a hyperspectral rice seed purity identification method based on the LASSO logistic regression model (LLRM). The feasibility of using LLRM for the selection of feature wavelength bands and seed purity identification are discussed using four types of rice seeds as research objects. The results of 13 different adulteration cases revealed that the value of the regularisation parameter was different in each case. The recognition accuracy of LLRM and average recognition accuracy were 91.67–100% and 98.47%, respectively. Furthermore, the recognition accuracy of full-band LRM was 71.60–100%. However, the average recognition accuracy was merely 89.63%. These results indicate that LLRM can select the feature wavelength bands stably and improve the recognition accuracy of rice seeds, demonstrating the feasibility of developing a hyperspectral technology with LLRM for seed purity identification.

## 1. Introduction

Paddy is one of the most important food sources worldwide, and its large number of varieties is continuing to increase. The phenomenon of seed mixing exists inevitably in each stage of paddy planting, production, and circulation, which affects the yield, nutrient composition, taste, and price of paddy rice. Therefore, the identification of rice seed purity is crucial in the grain industry. However, the identification task is primarily conducted manually with the naked eye [1] and can thus be easily affected by subjective factors such as experience and tiredness. Certain chemical or biological methods, such as phenol staining and protein electrophoresis, can accurately identify seed purity [2,3]; however, these methods are destructive, time-consuming, require large equipment or expertise, and cannot be batch processed. Therefore, it is vital to develop a rapid and accurate automatic identification method for rice seed purity.

Recently, nondestructive testing methods based on optical technology, such as infrared spectroscopy technology [4,5], machine vision, and hyperspectral technology [6,7,8,9,10,11,12,13], have received significant attention. Infrared spectroscopy technology identifies the spectral features of seeds [14] and analyses their internal chemical compositions (such as protein and starch) to classify them. However, this technology can only acquire a small area of spectral information from the seed samples at a time, and many repetitive measurements in different places are needed to represent the entire scenario of adulteration [15]. Machine vision technology captures colour RGB images of seeds and analyses their external physical features to classify them. Several reports on relevant machine vision technology have been published [16,17]. For example, Chaugule et al. (2014) classified four types of rice seeds based on their colour, shape, and texture with a recognition accuracy of 90.30–94.00% [18]. Additionally, they used the angle of seeds as a new identification feature in 2016, improving the identification accuracy to 97.6% [19]. In 2017, Huang et al. analysed the key details of both ends of rice seeds and found that the identification accuracies of three rice seeds with similar appearances were 92.68%, 97.35%, and 96.57%, respectively [20]. Furthermore, in 2021, Ansari et al. adopted machine vision technology combined with a multivariate analysis method to establish a rapid detection method of rice seeds based on different purities, obtaining the highest accuracy of 93.9% [21]. However, the phenomena of overlapping and superposition of external physical features are often observed owing to the similar appearances of many rice seeds. Therefore, it is difficult to obtain a stable classification effect using spatial information alone [22].

Hyperspectral technology can obtain the spectral and spatial information of seed samples, and it has attracted significant attention in the field of seed identification [23,24,25]. There have been many reports on the use of hyperspectral technology to identify corn, wheat, soybean, rice, and other grain particles. For example, in 2013, Silvia et al. used partial least squares discriminant analysis to identify hyperspectral images of groats and oats with an accuracy of approximately 100% [26]. In 2015, Yang et al. used a support vector machine (SVM) to identify waxy corn seeds, achieving the highest identification accuracy of 98.2% [27]. In 2020, Zhu et al. used a convolutional neural network to identify hyperspectral images of six types of soybean seeds with an average recognition rate of 91% [28]. The identification methods of rice can be broadly classified into two categories. The first incorporates the popular deep learning algorithms for classification and recognition. For example, in 2018, Qiu et al. used a convolutional neural network to identify four types of rice; they achieved a recognition rate of 87% using 12,000 training samples [29]. Furthermore, in 2020, Weng et al. used a deep learning network, namely principal component analysis (PCA) network, to train 4320 samples, obtaining the best recognition rate of over 98% for 10 rice varieties [30]. However, although the deep learning method does not require feature extraction, it requires a large number of training samples, leading to relatively high costs of hyperspectral image collection and data processing. The second category uses the traditional machine learning methods for identification. The most effective method is to use PCA dimension reduction and SVM classification. In 2015, Sun et al. used a combination of spectral, image, and textural features to identify four types of rice using SVM with a recognition accuracy of 91.6% [31]. Furthermore, in 2020, Liu et al. used a combination of spectral, image, and textural features to identify four types of rice seeds with an average identification accuracy of 98% [32]. These methods only require a small number of training samples to achieve a satisfactory recognition rate. However, hyperspectral spatial information contains a significant amount of redundant information and unpredictable noise, leading to high costs of hyperspectral image acquisition and processing. The cost of data acquisition and processing can be further reduced if the feature wavelength bands can be selected via some mathematical methods such as PCA and analysis of variance [33]. 

There are several methods to reduce the dimension of hyperspectral data [34,35], such as correlation analysis [36], beta coefficient of partial least squares regression analyses [37], and PCA [38]. PCA is widely used in the field of grain recognition [39,40,41,42]. However, it is essentially a dimensionality reduction method. Feature wavelength bands selected by PCA methods are not suitable for physical interpretation; it is easy to ignore those wavelength bands with a small contribution but contain more useful information for classification. Moreover, feature band selection and classification are two independent algorithms. However, there exists a complex interaction between feature band selection and classification modelling. The algorithm featuring simultaneous selection and classification is rarely used for the identification of rice seed purity.

In 1996, Tibshirani et al. proposed LASSO regression, a data sparse method [43] often used for hyperspectral data unmixing [44,45,46]. For the first time in 2017, Yang et al. introduced the LASSO method into hyperspectral images for band selection by conducting tests on public datasets. Band selection based on LASSO regression is more explanatory in physics and helps reveal the physical meaning [47]. The logistic regression model (LRM) is a probability classification method [48,49]. As a classifier, it is widely used in hyperspectral image processing, deep learning and other fields, such as medicine, social science, along with others [50,51,52,53]. This study combines the advantages of the sparse feature of the LASSO algorithm and the classification feature of the logistic regression model. Then, an identification method for hyperspectral rice seed purity based on the LASSO logistic regression model (LLRM) is proposed. We selected feature wavelength bands from four types of rice seeds according to different adulteration conditions. Thereafter, we established the prediction model and compared it with the full-band LRM. Finally, we discuss the feasibility of LLRM to select the feature bands and detect the purity of rice seeds.

## 2. Materials and Methods

### 2.1. Sample Preparation and Data Acquisition

We selected the following four types of rice seeds with plump grains: Huguangxiang, Xiangwan Japonica, Huanghuazhan, and Japonica 530. Each type contained 72 grains, which were naturally air-dried to be considered as experimental samples. Different types of seeds were neatly placed on the sample table with a spacing of approximately 5 mm to ensure that all the seeds had the same collection environment. Hyperspectral image data of all the seeds were collected four times owing to a large number of seeds.

A schematic of the hyperspectral imaging system for rice seeds is illustrated in Figure 1. The rice seed samples were placed directly on a standard whiteboard. Two bilinear light sources, including six halogen bulbs (H8, OSRAM light AG; Munich, Germany), were symmetrically placed on both sides of the whiteboard for illumination. Furthermore, the hyperspectral camera (Specim IQ, Specim, Spectral Imaging Ltd.; Oulu, Finland) was fixed on the top with a bracket [54]. Specim IQ is a handheld hyperspectral camera, which integrates a processor (NVIDIA Tegra K1, NVIDIA Corporation; Santa Clara, CA, USA), spectral camera (CMOS technology), viewfinder camera (5 Mpix), focus camera, and a scanner with the motor for optics movement. Its spectral resolution is 7 nm with 204 spectral bands across the wavelength range 400–1000 nm, and the image resolution is 512 × 512 px. The original data collected by the hyperspectral camera were light intensity information. These data need to be converted into reflectivity data for subsequent processing. The formula for reflection spectrum conversion is as follows:(1)$$R=\frac{D_{Raw}-D_{Dark}}{D_{White}-D_{Dark}},$$
where $D_{Raw}$, $D_{Dark}$, and $D_{White}$ denote the light intensity of the received sample, dark current intensity, and light intensity reflected by the whiteboard, respectively.

Figure 2 depicts the visible image of four rice seed samples obtained through the viewfinder of the hyperspectral camera; they are Huguangxiang, Xiangwan Japonica, Huanghuazhan, and Japonica 530 from the first (top) to fourth (bottom) rows, respectively. Huangxiang and Huanghuazhan are indica rice with a long and narrow shape, whereas Xiangwan Japonica and Japonica 530 are japonica rice with a short and wide shape. Although there are apparent differences in appearance between the japonica and indica rice, the shape features of the two japonica (or indica) varieties are significantly similar. Thus, it is difficult to identify the four varieties based on their image features alone owing to their biological diversity.

The original hyperspectral image also contained other objects, such as the reference whiteboard, in addition to the seed samples. We cut out the irrelevant parts to reduce the amount of data and save data processing resources, retaining only the region of interest (ROI). The original hyperspectral image contained 204 bands of information. The first and last few bands contained no useful information owing to significant noise. Therefore, only the data from the 15th to 190th bands (450–950 nm) were selected for subsequent processing.

### 2.2. Data Processing Flow

Figure 3 depicts the data processing flowchart of the hyperspectral rice seed purity identification technology. The first step was data pre-processing. The reflectivity data of the ROI were processed by standard normal variate (SNV) transformation and derivative methods. The second step was background segmentation, which involved segmenting the rice seed pixels from the background image and obtaining and analysing the spectral features of each rice seed. The third step involved preparing the training and the test set. The spectral features of 72 pure seeds of each type and the other three hybrid seeds were grouped together to simulate the purity of the seed, producing 13 datasets. These datasets were divided into training and test sets. The fourth step was modelling. LLRM was built based on the training set, and the selected spectral features were used to identify seeds. The last step was data output, wherein the accuracy of seed purity detection was determined based on the classification results.

### 2.3. Spectral Feature Extraction

#### 2.3.1. Data Pre-Processing

Various noises exist in the original hyperspectral data due to the limitations of the acquisition environment and the instrument, requiring that all data need to be pre-processed [55]. There are many methods suitable for pre-processing hyperspectral data, among which SNV and derivative methods are the most used. SNV is often used to eliminate the influence of sample surface scattering and optical path change on the spectrum [56,57]. Derivative processing includes first-order derivatives (FDs) and second-order derivatives (SDs) to eliminate other background interference and improve spectral resolution. Hence, this study uses SNV, FD, and SD methods to pre-process the data.

The SNV transformation formula is as follows:(2)$$S_{ij}\left(m\right)=\frac{x_{ij}\left(m\right)-{\bar{x}}_{ij}}{\sqrt{\frac{1}{N-1}\sum_{m=1}^{N}{\left(x_{ij}\left(m\right)-{\bar{x}}_{ij}\right)}^{2}}},$$
where $S_{ij}$ of $x_{ij}$ represents the spectral features before and after the transformation of the pixels in row i and column j, respectively. Furthermore, ${\bar{x}}_{ij}$ denotes the average reflectivity of $x_{ij}$, which is provided by: (3)$${\bar{x}}_{ij}=\frac{\sum_{m=1}^{N}x_{ij}\left(m\right)}{N}.$$

FDs and SDs are fulfilled by the Savitzky–Golay smoothing and differentiation filter [58,59], and the calculation formula as follows:(4)$$\begin{array}{c} dR\left(\lambda \right)=\frac{1}{110\Delta \lambda }\{5\left[R\left(\lambda +5\Delta \lambda \right)-R\left(\lambda -5\Delta \lambda \right)\right]+4\left[R\left(\lambda +4\Delta \lambda \right)-R\left(\lambda -4\Delta \lambda \right)\right]\\ -3\left[R\left(\lambda +3\Delta \lambda \right)-R\left(\lambda -3\Delta \lambda \right)\right]+2\left[R\left(\lambda +2\Delta \lambda \right)-R\left(\lambda -2\Delta \lambda \right)\right]\\ +\left[R\left(\lambda +\Delta \lambda \right)-R\left(\lambda -\Delta \lambda \right)\right]\}, \end{array}$$
(5)$$\begin{array}{c} d^{2}R\left(\lambda \right)=\frac{1}{429\Delta \lambda^{2}}\{15\left[R\left(\lambda +5\Delta \lambda \right)+R\left(\lambda -5\Delta \lambda \right)\right]+6\left[R\left(\lambda +4\Delta \lambda \right)+R\left(\lambda -4\Delta \lambda \right)\right]\\ -\left[R\left(\lambda +3\Delta \lambda \right)+R\left(\lambda -3\Delta \lambda \right)\right]-6\left[R\left(\lambda +2\Delta \lambda \right)+R\left(\lambda -2\Delta \lambda \right)\right]\\ -9\left[R\left(\lambda +\Delta \lambda \right)+R\left(\lambda -\Delta \lambda \right)\right]-10R(\lambda )\}, \end{array}$$
where the polynomial order is 2 and smoothing point is 11, and R(λ) represents the reflectance data for wavelength λ, and Δλ = 2.941 nm is the wavelength interval.

#### 2.3.2. Image Segmentation

The rice seed pixels must be separated from the background before the extraction of the spectral features. The traditional grains segmentation method selects a hyperspectral image of a certain wavelength band [60]. The adaptive threshold method was used to obtain the binarised image, which was used as a template. Thereafter, the hyperspectral data were multiplied to obtain image segmentation.

The traditional method only uses the spatial information of hyperspectral data, and it is difficult to obtain an accurate binarised image using only single wavelength band information due to the significant amount of noise in the hyperspectral data. However, the difference in spectral slope between the rice seeds and background was noticeable. The reflectance of rice seeds increases gradually in the 400–1000 nm region, and the spectral curve is relatively smooth, but the spectrum of the standard whiteboard is flat at 400–1000 nm. Therefore, we can use the slope information of the spectrum to synthesise a new greyscale image. The formula of a synthesised greyscale image is as follows:(6)$$g_{ij}=arctan\left(\frac{min\left(x_{ij}\right)}{max\left(x_{ij}\right)}\right),$$
where $g_{ij}$ represents the synthesised grey value of the pixel at the ith row and jth column, and $x_{ij}$ denotes the spectrum of pixel points in the ith row and jth column. The arctan function ensures that the value remains between 0 and 1. Thereafter, the synthesised greyscale image was binarised by Otsu’s method to obtain a more accurate sample image boundary. This method uses spectral slope as an effective discriminator, which is equivalent to using both the spatial and spectral information of hyperspectral data to fulfil image segmentation, so it has a better segmentation effect.

Figure 4 depicts the effect of the image segmentation after using the adaptive threshold method to binarise the synthesised greyscale image. Figure 4a depicts a colour image of a rice seed with a cast shadow due to an illumination problem, significantly affecting the image segmentation. Figure 4b depicts the binarised image using a hyperspectral image at 601.55 nm. Figure 4c showcases the result of segmentation using 601.55 nm wavelength band information. Some background pixels are incorrectly segmented due to the influence of the shadow, thereby affecting the spectral features of the rice seeds. Figure 4d shows the synthesised greyscale image using spectral slope. The spectral information helps overcome the effect of the shadow. Figure 4e represents the binarised spectral slope greyscale image. Figure 4f depicts the segmentation result using spectral slope information. Thus, the boundary of the rice seeds can be accurately determined using spectral slope information, yielding a considerably better result than that depicted in Figure 4c.

#### 2.3.3. Spectral Feature

The pixels from each rice seed formed a connected region after the rice seeds were segmented from the background. Thereafter, all the connected areas in the image were counted. The pixels in each connected area belonged to the same rice seed sample. The spectral features of each rice seed were obtained by averaging the spectral data of the same connected area. Figure 5 depicts the average spectral features of four types of rice seeds. Figure 5a illustrates the reflectance spectrum after correction. Figure 5b depicts the spectrum after SNV processes. The average spectral differences among different paddy varieties were not significant for both raw and SNV data. Figure 5c,d show some differences in the spectra of different varieties after the FD and SD processes. Therefore, it is necessary to establish a mathematical classification model to distinguish these bands.

### 2.4. LLRM Foundation and Optimisation

#### 2.4.1. Data Grouping

The spectral features of 72 pure seeds of each seed type and the other three adulterated seeds were grouped to simulate different cases of seed purity detection, producing 13 datasets. Table 1 presents the detailed information. Groups 1–10 considered Huguangxiang as the research object. Groups 1–4 were adulterated with the other three types of seeds in equal amounts. However, their proportions were different. Groups 5–7 comprised a mix of the other three seeds in different amounts. Groups 8–10 were adulterated with one other seed. Groups 11–13 considered Xiangwan Japonica, Huanghuazhan, and Japonica 530 as the research objects, respectively. Each group was adulterated with three other seeds, and each of them had 12 grains.

#### 2.4.2. Partition of Sample Sets

Sample set partitioning based on joint x-y distance (SPXY) algorithm was used to randomly divide the data into training and test sets, increasing the difference and representativeness between the samples and improving the stability of the model sets [61,62]. The formula to calculate the distance between two samples is as follows:(7)$$d_{xy}\left(p,q\right)=\frac{d_{x}\left(p,q\right)}{max\left(d_{x}\left(p,q\right)\right)}+\frac{d_{y}\left(p,q\right)}{max\left(d_{y}\left(p,q\right)\right)},$$
where
(8)$$d_{x}\left(p,q\right)=\sqrt{\sum_{k=1}^{M}{\left[x_{p}\left(k\right)-x_{q}\left(k\right)\right]}^{2}}\;\mathrm{and}$$
(9)$$d_{y}\left(p,q\right)=\left|y_{p}-y_{q}\right|.$$

Presented here, (p,q)∈[1,N], p and q denote the numbers of two different samples, $x_{p}$ and $x_{q}$ represent the spectral features of samples p and q, respectively. Furthermore, $d_{x}$ and $d_{y}$ denote the spectral distance and category distance of two different samples, respectively.

#### 2.4.3. LLRM

The identification of rice seed purity can be considered a binary classification problem because it does not require the identification of non-target seed varieties. Let us assume that N represents the number of samples of rice seeds with a P-dimensional feature. Thus, the input feature, *X*, is an N×P matrix, and the output $Y=\left[y_{1},y_{2},\cdots y_{N}\right]$ is a binary variable. If the sample is of the target seed variety, $y_{i}=1$; else, $y_{i}=0$. The logarithmic probability of the ith sample belonging to target seed variety is:(10)$$log\left\{\frac{p\left(y_{i}=1|X^{i},\beta \right)}{1-p\left(y_{i}=1|X^{i},\beta \right)}\right\}=\eta_{\beta }\left(X^{i}\right),$$
where $X^{i}=\left[x_{i1},x_{i2},\cdots x_{iP}\right]$ represents the feature of the ith sample, $\beta =\left[\beta_{0},\beta_{1},\beta_{2},\cdots \beta_{P}\right]$ denotes the regression coefficient of LLRM, and $p\left(y_{i}=1|X^{i},\beta \right)$ represents the probability of the ith sample, which belongs to the target seed variety when the regression coefficient is β.
(11)$$\eta_{\beta }\left(X^{i}\right)=\beta_{0}+\beta_{1}x_{i1}+\beta_{2}x_{i2}+\cdots \beta_{p}x_{iP}=\sum_{j=0}^{P}\beta_{j}x_{ij}.$$

The logarithmic maximum likelihood function of LLRM is as follows:(12)$$l\left(\beta \right)=\sum_{i=1}^{N}\left\{y_{i}\eta_{\beta }\left(X^{i}\right)-log\left\{1+exp\left[\eta_{\beta }\left(X^{i}\right)\right]\right\}\right\}.$$

LASSO regression minimises the sum of squares of residuals when the sum of absolute values of regression coefficients is lesser than a normal constraint. Some regression coefficients were gradually compressed during the parameter estimation; when the regression coefficient was lower than the threshold, it was set to 0 to achieve variable screening. The L1 norm was introduced into LLRM as the punishment term. Furthermore, the objective of variable sparsity was achieved. Its regression cost function is as follows:(13)$$S_{\lambda }\left(\beta \right)=-l\left(\beta \right)+\lambda {\left\|\beta \right\|}_{1}.$$

The estimated regression coefficient, $\hat{\beta }\left(\lambda \right)$, was obtained by minimising Equation (7) in the model.
(14)$$\hat{\beta }\left(\lambda \right)=\begin{array}{c} argmin\\ \beta \end{array}\sum_{i=1}^{N}\left\{y_{i}\eta_{\beta }\left(X^{i}\right)-log\left\{1+\exp\left[\eta_{\beta }\left(X^{i}\right)\right]\right\}\right\}+\lambda \sum_{j=1}^{P}\left|\beta_{j}\right|,$$
where λ represents the regularisation parameter. The smaller the value of λ, the smaller the punishment, retaining more variables. However, the larger the value of λ, the larger the punishment, retaining fewer variables. Therefore, the selection of an appropriate λ has a significant impact on the classification accuracy of LLRM. This study uses a ten-fold cross-validation method to select the optimal value of λ and can be calculated as follows:(15)$${\hat{\lambda }}_{CV}=\begin{array}{c} argmin\\ \lambda >0 \end{array}K^{-1}CV_{k}\left(\lambda \right),$$
where K=10, $CV_{k}\left(\lambda \right)$ represents an error in fitting the *k*th subset when the optimal value of λ is used.

The detailed steps of the LLRM algorithm implementation are as follows:(1)Data pre-processing: The data collected by hyperspectral technology were pre-processed. Then, they were grouped, as presented in Table 1.(2)Dividing the training and test sets: The SPXY algorithm was used to randomly divide the data into training and test sets with a 1:1 ratio.(3)Selecting the regularisation parameter: The training samples were divided into 10 parts using the cross-validation method, taking turns using 9 of their train, and then calculating the error of fitting the other one. The mean square error of the 10 prediction results was used to estimate the accuracy of the algorithm. Moreover, the optimal value of λ was selected based on the accuracy of the model prediction and by combining the number of feature bands. (4)Selecting the feature wavelength bands: The coordinate descent method was used to calculate the regression coefficient at an optimal value of λ. Then, the feature band was selected based on the regression coefficient.(5)Modelling: The feature wavelength bands from the test set were selected. Moreover, the regression model was considered the foundation.(6)Calculation of classification accuracy: The function is calculated as follows:(16)$$p\left(i\right)=1/\left(1+exp\left(-y_{i}\right)\right).$$

If the probability is greater than 0.5, the sample represents the target seed variety. However, if the probability is less than 0.5, the sample represents a non-target seed variety. Finally, they were compared with the labels and the classification accuracy was determined.

#### 2.4.4. Optimum λ Selection

The value of λ has a considerable influence in solving the LLRM. An optimum value of λ can establish a balance between the fitting error and model complexity. If λ is significantly large, the degree of model compression increases, the number of selected feature wavelength bands decreases, and the fitting error increases, consequently affecting classification. If λ is significantly small, the variable compression effect is constrained, and the number of selected feature bands increases significantly, wasting computing resources. Therefore, the value of λ must be optimised carefully.

According to the number of feature wavelengths and recognition accuracy, the optimisation range of λ is set to 0.0001–0.2 in this study. We used cross-validation to determine the relatively small range of λ [63,64]. Thereafter, the number of feature bands and the accuracies for different λ values were analysed, and the optimal λ value was selected. Figure 6a depicts the change in the regression coefficient with λ. The relative value of the regression coefficient gradually decreases, and the coefficient is gradually compressed with increasing λ. Figure 6b depicts variation in the data fitting error of Group 2 with the change in λ. The blue curve represents the fitting error, and the thick red line indicates the standard deviation of different fitting errors. The red vertical line corresponds to the value of λ with the smallest fitting error. In general, the smallest fitting error does not imply the best prediction effect because the data contain unknown noise. Therefore, the optimal value of λ was expected to be found near the red vertical line. Figure 6c,d show that the number of feature wavelength bands is 10 when the fitting error is the smallest with a prediction accuracy of 92.57%. However, the prediction accuracy does not decrease with the decreasing number of feature wavelength bands. This indicates the need to further optimise the model. The highest prediction accuracy of 95.37% is obtained below the red vertical line. There are two positions with the highest prediction rate of λ = 0.0270 and λ = 0.0668, which correspond to 10 and 7 feature wavelength bands, respectively. Moreover, the green vertical line corresponds to λ = 0.0668, which requires the minimum number of feature wavelength bands while exhibiting the same prediction accuracy, thereby effectively reducing the cost of data processing. Other data groups can be optimised similarly to obtain the best LLRM.

## 3. Results

### 3.1. Results of Wavelength Bands Selection

Table 2 presents the wavelength bands selected using the SNV data under different adulteration groups. Groups 1–10 indicate that the coincidence degree of the selected wavelength bands is significantly higher when the same variety of rice seeds is considered despite the different number and variety of doped seeds. Among them, wavelength bands of 678.71 nm, 810.86 nm, and 871.60 nm appeared in all the data groups. Furthermore, the appearance frequencies of 548.55 nm, 798.77 nm, and 847.25 nm were also significantly higher, indicating the feasibility of using LLRM for band selection. Groups 11–13 show that the coincidence degree of the selected wavelength bands is significantly smaller, implying that different LLRMs must be established for different rice varieties.

Table 3 and Table 4 list the selected wavelength bands using FD and SD data under different adulteration groups, respectively. The coincidence degree of the selected wavelength bands is significantly higher for the same variety of rice seeds. In contrast, it is significantly smaller for different varieties of rice seeds. We compared the results of band selection using SNV, FD, and SD data for the same type of target seeds. The results indicate that the band coincidence is extremely low, implying that the band selection of LLRM is highly sensitive to the data pre-processing procedure. Under the optimal LLRM, the number of feature wavelength bands selected by the SNV, FD, and SD data were 5–17, 3–11, and 6–12, respectively. Moreover, the FD data required the least number of bands among the three.

### 3.2. Band Selection

Figure 7 depicts the wavelength band selection using the SNV data for different rice varieties. The red lines indicate the average spectra of the target rice seeds, and the blue lines represent the average spectra of the adulterated seeds. The positions indicated by the green lines represent the best bands based on the data of Group 2. Figure 7a shows that the wavelength bands of 548.55 nm, 714.55 nm, and 810.86 nm correspond to small reflectance peaks of the target seeds. The reflectance of the target seed spectra at wavelengths of 643.01 nm, 678.71 nm, and 871.60 nm are significantly different from those of the adulterated seeds. The difference in spectra between the target rice seeds and adulterated seeds are not noticeable at 762.98 nm, 798.77 nm, and 847.25 nm. Certain wavelength bands, such as 896.01 nm and 920.48 nm, are not selected despite the noticeable difference in their spectra, implying that greater spectral differences do not contain more useful information for classification. The number and location of the selected bands in Figure 7b–d are significantly different, indicating that different LLRMs must be established for different varieties.

Figure 8 depicts the wavelength band selection using FD data for different rice varieties. Furthermore, Figure 8a shows that there are significant differences in the data at 466.77 nm, 583.85 nm, 678.71 nm, and 835.11 nm. The greater spectral differences do not imply a greater variety of samples. Comparing Figure 8b–d, we find that the selected wavelength bands are mainly concentrated in the infrared wavelengths of 750–900 nm. Moreover, the spectral differences cannot be observed directly in the hyperspectral data, indicating that the infrared bands contain more useful information than visible light bands. Thus, we conclude that more attention should be allotted to the infrared bands for wavelength band selection.

Figure 9 depicts the wavelength band selection using SD data for different rice varieties. It is difficult to notice the spectral differences in different wavelength bands because the SD data is significantly rough. Hence, the wavelengths with large peaks are not selected. Comparing Figure 7, Figure 8 and Figure 9, we find that the selected feature wavelength bands are almost different under different pre-processing conditions for the same type of target seeds, indicating that the data pre-processing method significantly influences the LLRM. The pre-processing using the FD data exhibited the fastest convergence speed and required the least number of bands; hence, it is the best pre-processing method.

The coincidence degrees of the feature bands selected for different data groups were significantly higher in the Huguangxiang case. The appearance frequencies of 548.55 nm, 643.01 nm, 678.71 nm, 714.55 nm, 810.86 nm, and 871.60 nm were the highest. The coincidence degrees of the feature bands were significantly smaller for different types of rice seeds, indicating that a specific LLRM must be established for different rice varieties. The band selection results of the LLRM were highly sensitive to data pre-processing. It was found that pre-processing with FD was more suitable for LLRM than with SNV or SD.

### 3.3. Comparison of Model Accuracy

Table 5, Table 6 and Table 7 show the recognition accuracies between LLRM and LRM using SNV, FD an SD data, respectively. As shown in Table 5, the recognition accuracy of LLRM is 98.77–100%, with an average recognition accuracy of 99.64%. The recognition accuracy of the full variable LRM is 71.60–100%, with an average recognition rate of 93.66%. However, the results of LRM are considerably lower than those of LLRM. The recognition accuracies of Group 3 and 4 are particularly more prominent at less than 80%; this indicates that LLRM selects the key bands steadily and eliminates unnecessary noise, thereby improving the signal-to-noise ratio of the hyperspectral data and obtaining a better rice seed recognition rate.

Table 6 presents the comparison of the recognition accuracies between LLRM and LRM using FD data. The recognition accuracy of LLRM is 95.37–100%, with an average recognition accuracy of 99.18%. However, the recognition accuracy of the full variable LRM is 73.33–95.83%, with an average recognition rate of merely 88.63%. Table 7 shows that the recognition accuracy of the full variable LRM using SD data is 77.78–95.61%, with an average recognition rate of merely 86.60%. However, the recognition accuracy of LLRM is 91.67–100%, with an average recognition rate of 96.59%. The average recognition rate of LLRM is higher than the highest rate of LRM. Comparing Table 5 with Table 6 and Table 7 shows that the recognition accuracy of the full variable LRM is 78.89–95.61%, with an average recognition rate of merely 86.60%. However, the recognition accuracy of LLRM is 91.67–100%, with an average recognition rate of 96.59%. This result indicates that it is feasible to develop a hyperspectral technology with LLRM to identify seed purity. 

## 4. Conclusions

In this study, we combined the sparse characteristics of the LASSO algorithm and the classification characteristics of the LRM and proposed a method to identify the hyperspectral rice seed purity based on the LLRM. We tested 13 different adulterated groups of four varieties of rice seeds. The results of different data groups indicated that the value of the regularisation parameter λ was different for different adulteration cases. Moreover, the number of selected feature wavelength bands also varied. The best balance between the number of feature wavelength bands and recognition accuracy was obtained by adjusting λ, thereby establishing the optimal LLRM.

The recognition accuracy of the LLRM was 91.67–100% in 13 rice seed adulteration cases, with an average recognition rate of 98.47%. However, the recognition accuracy of the LRM was 71.60–100%, with an average accuracy of merely 89.63%. This result indicates that LLRM selects the key bands steadily and improves the signal-to-noise ratio of the hyperspectral data, obtaining better accuracy in rice seed recognition. The average number of selected feature wavelength bands is 11.3 for SNV data, 8.5 for FD data, and 9.5 for SD data. SNV data selected relatively more feature wavelength bands, which led to better accuracy than FD and SD data. By comparing all the LLRMs, we found that the accuracy of three pre-processing methods was improved by 5.98%, 10.55%, and 9.99%, respectively, and the probability of achieving a 100% recognition accuracy increased after FD processing, implying that FD processing is more suitable than other approaches for LLRM.

These results reveal that the hyperspectral detection technology of rice seed purity based on LLRM can reduce the collection cost and improve the processing speed while ensuring satisfactory recognition accuracy. Thus, LLRM demonstrates significant potential for application in the field of seed purity identification. In future research, we will further analyse and select these candidate feature wavelength bands and add information on the texture and shape of samples to establish a more robust hyperspectral rice seed purity identification technology.

## Figures and Tables

**Figure 1 sensors-21-04384-f001:**
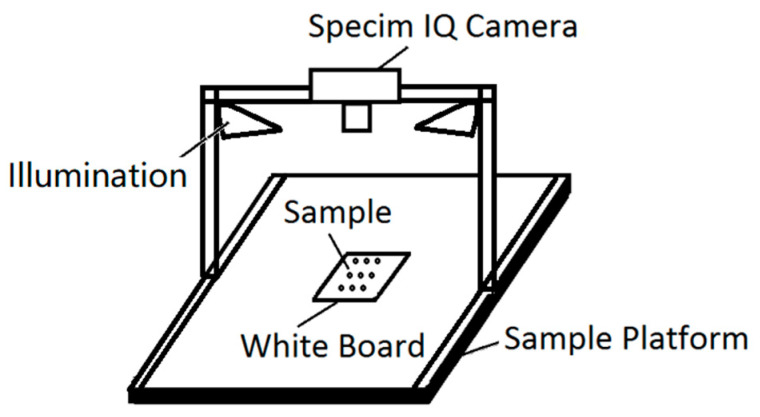
Schematic of the hyperspectral imaging system for rice seeds.

**Figure 2 sensors-21-04384-f002:**
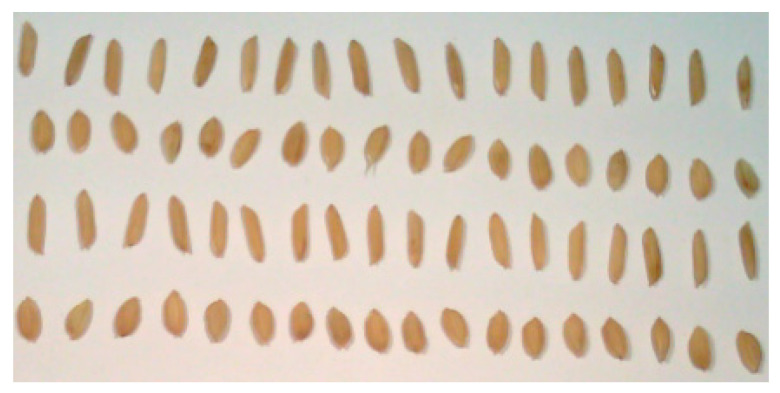
Image of four rice seed samples through the viewfinder of the hyperspectral camera; the first (top) to fourth (bottom) rows are Huguangxiang, Xiangwan Japonica, Huanghuazhan, and Japonica 530, respectively.

**Figure 3 sensors-21-04384-f003:**
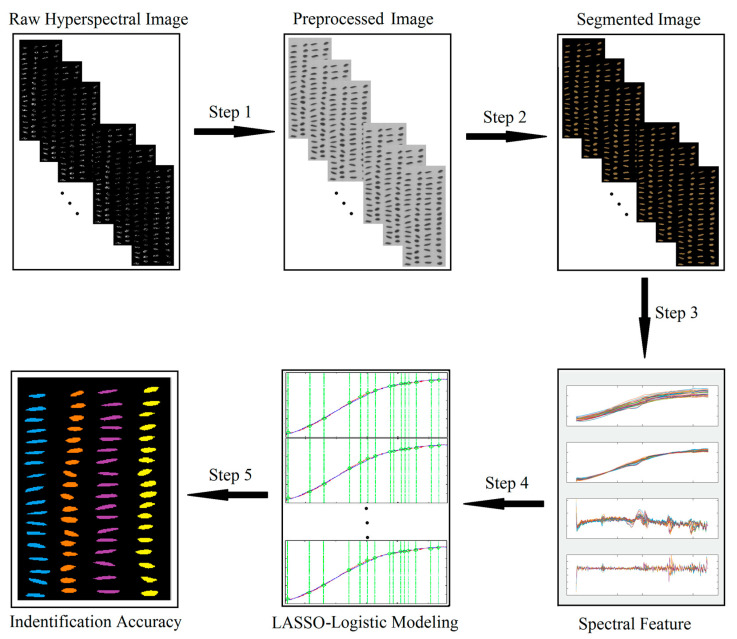
Flowchart of hyperspectral image processing, including data pre-processing, background segmentation, data set preparation, modelling, and identification accuracy output.

**Figure 4 sensors-21-04384-f004:**
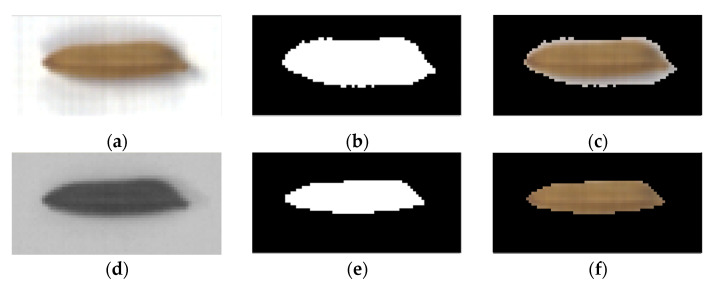
Example of background segmentation using synthesised spectral slope greyscale image. (**a**) Colour image of a seed; (**b**) Binarised image using single-band (601.55 nm) information; (**c**) Segmentation result using single-band (601.55 nm) information; (**d**) Synthesised spectral slope greyscale image; (**e**) Binarised image using spectral slope information; (**f**) Segmentation result using spectral slope information.

**Figure 5 sensors-21-04384-f005:**
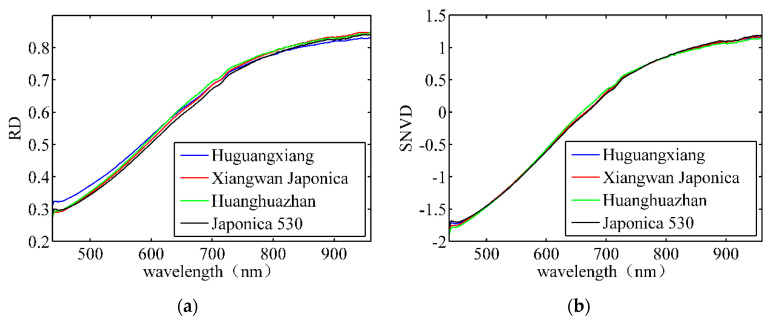
Average spectral features of four rice seeds. (**a**) Original reflectance spectrum; (**b**) spectrum after SNV; (**c**) spectrum after FD’ (**d**) spectrum after SD.

**Figure 6 sensors-21-04384-f006:**
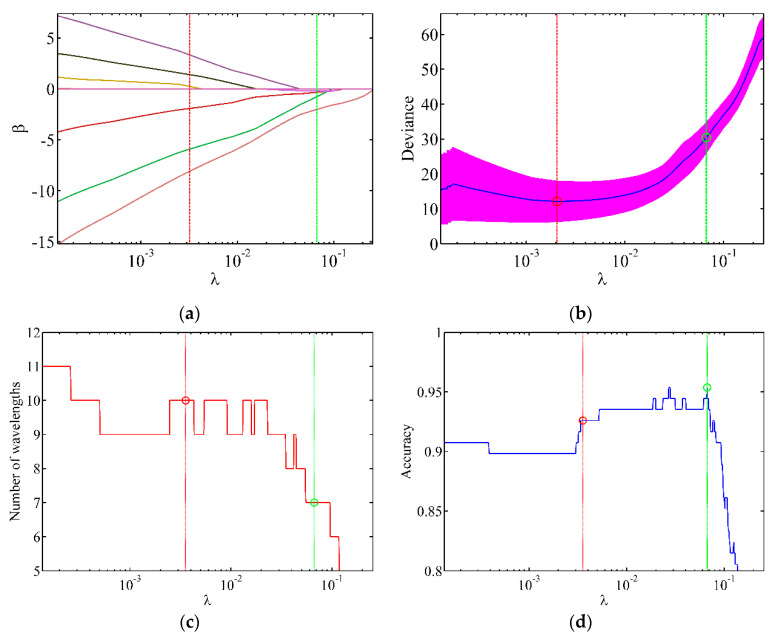
Change in parameters with different λ values. (**a**) Regression coefficient; (**b**) fitting deviation; (**c**) number of feature wavelength bands; (**d**) prediction accuracy.

**Figure 7 sensors-21-04384-f007:**
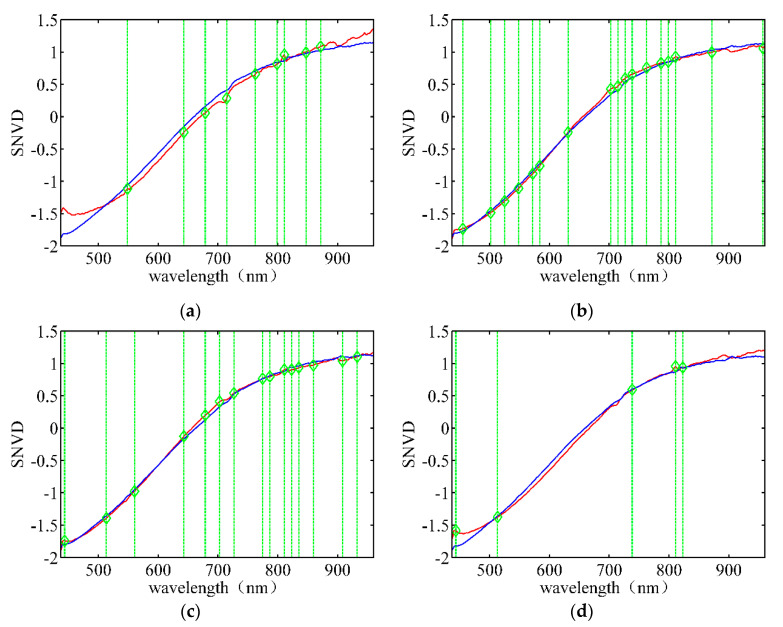
Wavelength band selection using SNV data for different rice varieties under the optimal λ. (**a**) Huguangxiang; (**b**) Xiangwan Japonica; (**c**) Huanghuazhan; (**d**) Japonica 530.

**Figure 8 sensors-21-04384-f008:**
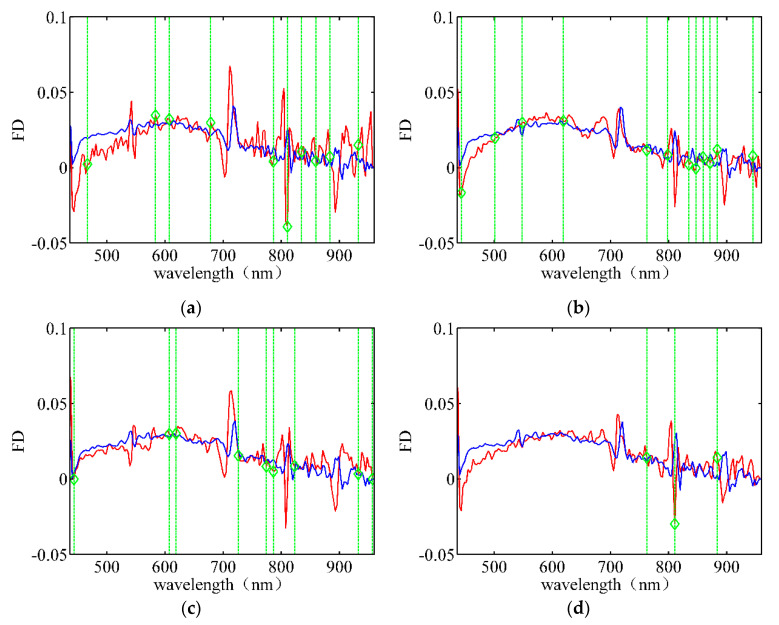
Wavelength band selection using FD data for different rice varieties under the optimal λ. (**a**) Huguangxiang. (**b**) Xiangwan Japonica. (**c**) Huanghuazhan. (**d**) Japonica 530.

**Figure 9 sensors-21-04384-f009:**
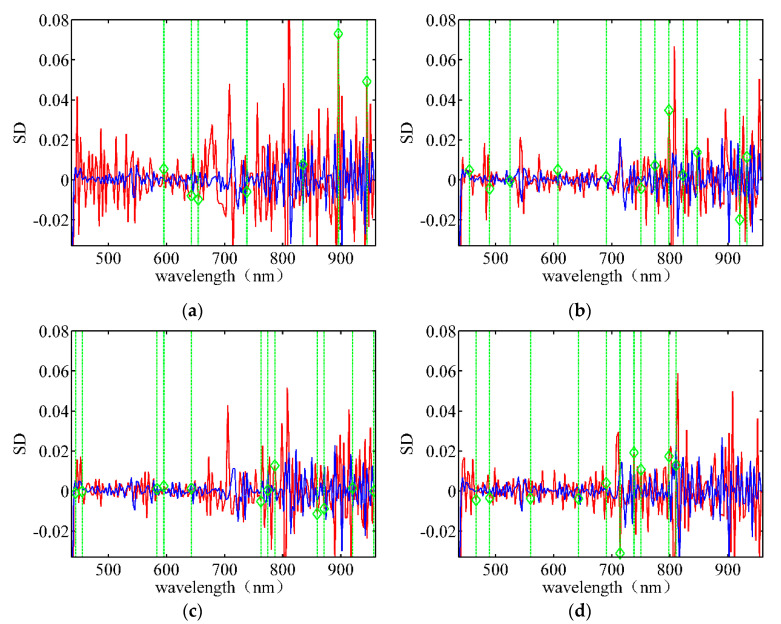
Wavelength band selection using SD data for different rice varieties under the optimal λ. (**a**) Huguangxiang; (**b**) Xiangwan Japonica; (**c**) Huanghuazhan; (**d**) Japonica 530.

**Table 1 sensors-21-04384-t001:** Data grouping details to simulate different cases of seed purity detection.

Unit/Grain	Type 1	Type 2	Type 3	Type 4	Sum
Group 1	72	24	24	24	144
Group 2	72	12	12	12	108
Group 3	72	6	6	6	90
Group 4	72	3	3	3	81
Group 5	72	24	12	6	114
Group 6	72	12	24	6	114
Group 7	72	6	12	24	114
Group 8	72	36	0	0	108
Group 9	72	0	36	0	108
Group 10	72	0	0	36	108
Group 11	12	72	12	12	108
Group 12	12	12	72	12	108
Group 13	12	12	12	72	108

**Table 2 sensors-21-04384-t002:** The results of feature wavelength bands selected using SNV data under different adulteration groups.

Group	Feature Wavelength Bands [nm]	Number
Group 1	548.55, 572.07, 643.01, 678.71, 714.55, 762.57, 798.77, 810.86, 822.98, 847.25, 871.60, 896.01, 908.24, 920.48	14
Group 2	548.55, 643.01, 678.71, 714.55, 762.57, 798.77, 810.86, 847.25, 871.60	9
Group 3	548.55, 631.15, 643.01, 678.71, 762.57, 798.77, 810.86, 847.25, 871.60, 920.48	10
Group 4	548.55, 619.30, 678.71, 714.55, 798.77, 810.86, 871.60	7
Group 5	548.55, 572.07, 583.85, 643.01, 678.71, 714.55, 762.57, 786.68, 810.86, 822.98, 847.25, 871.60, 896.01, 920.48	14
Group 6	536.82, 548.55, 619.30, 678.71, 714.55, 762.57, 798.77, 810.86, 847.25 871.60, 896.01	11
Group 7	548.55, 643.01, 678.71, 714.55, 750.54, 762.57, 798.77, 810.86, 847.25, 871.60, 957.32	11
Group 8	548.55, 560.30, 572.07, 631.15, 643.01, 678.71, 762.57, 810.86, 847.25, 871.60, 896.01, 908.24, 920.48	13
Group 9	525.10, 536.82, 619.30, 678.71, 726.53, 786.68, 798.77, 810.86, 847.25, 871.60, 896.01	11
Group 10	560.30, 643.01, 678.71, 714.55, 798.77, 810.86, 871.60, 908.24, 920.48, 957.32	10
Group 11	455.16, 501.72, 525.10, 548.55, 572.07, 583.85, 631.15, 702.58, 714.55, 726.53, 738.53, 762.57, 786.68, 798.77, 810.86, 871.60, 957.32	17
Group 12	443.56, 513.40, 560.30, 643.01, 678.71, 702.58, 726.53, 774.62, 786.68, 810.86, 822.98, 835.11, 859.42, 908.24, 932.74	15
Group 13	443.56, 513.40, 738.53, 810.86, 822.98	5

**Table 3 sensors-21-04384-t003:** The results of feature wavelength bands selected using FD data under different adulteration groups.

Group	Feature Wavelength Bands [nm]	Number
Group 1	466.77, 583.85, 607.46, 678.71, 750.54, 822.98, 835.11, 847.25, 859.42, 883.79, 932.74	11
Group 2	466.77, 583.85, 607.46, 678.71, 786.68, 810.86, 835.11, 859.42, 883.79, 932.74	10
Group 3	455.16, 583.85, 666.79, 678.71, 810.86, 835.11, 859.42, 871.60, 883.79, 932.74	10
Group 4	466.77, 583.85, 678.71, 810.86, 835.11, 859.42, 883.79	7
Group 5	466.77, 583.85, 678.71, 835.11, 859.42, 883.79	6
Group 6	466.77, 654.89, 714.55, 822.98, 859.42, 883.79, 932.74	7
Group 7	455.16, 466.77, 525.10, 548.55, 607.46, 678.71, 786.68, 835.11, 859.42, 883.79, 932.74	11
Group 8	466.77, 583.85, 678.71, 835.11, 859.42, 883.79	6
Group 9	466.77, 501.72, 619.30, 702.58, 714.55, 822.98, 835.11, 847.25, 859.42, 883.79	10
Group 10	455.16, 525.10, 607.46, 786.68, 835.11, 847.25, 859.42, 883.79, 932.74	9
Group 11	443.56, 501.72, 548.55, 619.30, 762.57, 798.77, 835.11, 847.25, 859.42, 871.60, 883.79, 945.02	12
Group 12	443.56, 607.46, 619.30, 726.53, 774.62, 786.68, 822.98, 932.74, 957.32	9
Group 13	762.57, 810.86, 883.79	3

**Table 4 sensors-21-04384-t004:** The results of feature wavelength bands selected using SD data under different adulteration groups.

Group	Feature Wavelength Bands [nm]	Number
Group 1	478.41, 560.30, 619.30, 643.01, 654.89, 690.64, 726.53, 786.68, 822.98, 847.25, 896.01	11
Group 2	595.65, 643.01, 654.89, 738.53, 835.11, 896.01, 945.02	7
Group 3	595.65, 643.01, 654.89, 726.53, 738.53, 835.11, 859.42, 945.02	8
Group 4	466.77, 643.01, 654.89, 726.53, 738.53, 774.62, 835.11, 847.25, 859.42, 945.02	10
Group 5	631.35, 643.01, 654.89, 690.64, 726.53, 738.53, 786.68, 847.25, 896.01	9
Group 6	643.01, 654.89, 690.64, 738.53, 786.68, 847.25, 896.01, 945.02	8
Group 7	478.41, 560.30, 643.01, 654.89, 690.64, 702.58, 738.53, 786.68, 822.98, 847.25, 896.01, 945.02	12
Group 8	643.01, 654.89, 690.64, 726.53, 738.53, 786.68, 822.98, 835.11, 847.25, 896.01	10
Group 9	455.16, 560.30, 643.01, 654.89, 690.64, 786.68, 896.01, 945.02	8
Group 10	654.89, 690.64, 702.58, 822.98, 883.79, 896.01	6
Group 11	455.16, 490.06, 525.10, 607.46, 690.64, 750.54, 774.62, 798.77, 822.98, 847.25, 920.48, 932.74	12
Group 12	443.56, 455.16, 583.85, 595.65, 643.01, 762.57, 774.62, 786.68, 859.42, 871.60, 920.48, 957.32	12
Group 13	466.77, 490.06, 560.30, 643.01, 690.64, 714.55, 738.53, 750.54, 798.77, 810.86	10

**Table 5 sensors-21-04384-t005:** Comparison of recognition accuracy between LLRM and LRM using SNV data.

	**Group 1**	**Group 2**	**Group 3**	**Group 4**	**Group 5**	**Group 6**	**Group 7**
LLRM	99.31%	100%	100%	98.77%	100%	100%	100%
LRM	97.22%	97.22%	78.89%	71.60%	100%	97.37%	95.61%
	**Group 8**	**Group 9**	**Group 10**	**Group 11**	**Group 12**	**Group 13**	**Average**
LLRM	99.07%	100%	100%	99.07%	99.07%	100%	99.64%
LRM	96.30%	96.30%	100%	94.44%	97.22%	95.37%	93.66%

**Table 6 sensors-21-04384-t006:** Comparison of recognition accuracy between LLRM and LRM using FD data.

	**Group 1**	**Group 2**	**Group 3**	**Group 4**	**Group 5**	**Group 6**	**Group 7**
LLRM	98.61%	100%	100%	100%	100%	100%	99.12%
LRM	95.83%	86.11%	73.33%	74.07%	87.72%	90.35%	94.74%
	**Group 8**	**Group 9**	**Group 10**	**Group 11**	**Group 12**	**Group 13**	**Average**
LLRM	100%	100%	99.07%	95.37%	97.22%	100%	99.18%
LRM	92.59%	87.96%	97.22%	88.89%	84.26%	99.07%	88.63%

**Table 7 sensors-21-04384-t007:** Comparison of recognition accuracy between LLRM and LRM using SD data.

	**Group 1**	**Group 2**	**Group 3**	**Group 4**	**Group 5**	**Group 6**	**Group 7**
LLRM	91.67%	95.37%	96.67%	98.77%	93.86%	92.98%	94.74%
LRM	88.19%	81.48%	78.89%	77.78%	80.70%	83.33%	95.61%
	**Group 8**	**Group 9**	**Group 10**	**Group 11**	**Group 12**	**Group 13**	**Average**
LLRM	100%	97.22%	100%	98.15%	96.30%	100%	96.59%
LRM	86.11%	87.96%	94.40%	91.67%	84.26%	95.37%	86.60%

## Data Availability

The data that support the findings of this study are available upon request from the authors.

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
