# Peer review of "Rice Seed Purity Identification Technology Using Hyperspectral Image with LASSO Logistic Regression Model"

_sensors, 2021, doi:10.3390/s21134384_

Round 1

Reviewer 1 Report

In this study 4 different types of paddy rice seeds were classified using a Specim IQ hyperspectral imaging sensor. The proposed classification algorithm combines LASSO and LRM methodologies with relatively high identification success rates.

General comments:

  1. The authors use often the phrase  "image information" (e.g. lines 12, 86 etc), while I believe they mean spatial information? Please revise.
  2. Do you have a ny reference for the statement in lines 46-49?
  3. lines 65-66, chemical composition can only be acquired after data analysis I believe.
  4. lines 92-94, can you be a little more specific regarding the statement here? For example refer some mathematical methods, like PCA etc.
  5. lines 100-102, what do you mean by "sample difference information"?
  6. Figure 2: what would be the outcome if you would just mix the seeds instead of having the same seeds in a row?
  7. repetition in line 163, "the test set".
  8. Any references on the statement in lines 175-176?
  9. How do you end up with the equation 4?
  10. line 213: can you please elaborate more on how you used the spectral slope calculation? It is no clear how you got this applied. Regarding the binirisation, did you tried Otsu's method?
  11. Table and figure captions should be more detailed and descriptive.
  12. line 275: how do you define which will be set to zero?
  13. At some points the text has some vague statements where the point is not clear. Please revise and provide more details (major issues were noted in my comments above).

Thank you.

Reviewer 2 Report

The paper “Rice Seed Purity Identification Technology Using Hyperspectral Image With LASSO Logistic Regression Model” is an interesting example of hyperspectral imaging techniques and an identification method based on regression model. 
This manuscript is admitted to the scope of the journal and well presented, however, just in case, requires some refinements and revisions to be more clear and convincing. 
Here are some minor fixes at specific points:

Line 129, 131: This manuscript missed information about the used equipment for the experiments such as whiteboard, two halogen lamps, and handheld hyperspectral camera. Read a couple of accepted journals then correct them according to the previous ones. 
Line 160: SNV stands for standard normal variate instead of standard normal variable, in general. 

From Introduction section, logistic regression model (LRM) has no citation. LRM is the only detection and indication method for rice kernels in this manuscript so some citations are required. Moreover, add up more LRM applications to the Introduction section to help readers to understand the manuscript. 

Line 186: Second-order derivative is introduced in Equation 4, however, first-order derivative isn’t. Besides, Equation 4 is difficult to understand due to lack of information. What is 1/429 and 15, 6, -1, -9 and 10 in the equation. A more detailed explanation is needed for the reader to understand. The revised sentences should be added up to the manuscript. 
Line 197-198: Difference in spectral slope method was used to make a proper mask for rice kernel. And, the index g was used to show the calculated synthesized greyscale image. To make sure, In Fig. 4 (d), scale bar is necessary to figure it out the synthesized greyscale image has proper index between 1 and 100 as mentioned line 205. And how to get the single band for making mask? I believe there are more than 15 methods for making binary image, why the greyscale image was selected to make a mask image?
Line 251-260: Section 2.4.2. introduces the method of partition of sample sets based on joint x-y distance algorithm that is SPXY. A few proper citations can provide good information about SPXY. 
Line 321: How did you get the range of initial value of lambda was 0.0001-0.2? Besides, a few proper citations can make help readers to understand issues to find the optimum lambda value. 
Table 5-7: Three groups 2,3,4 and 5 (relatively small number of samples as well 12, 6, 3 and 6 to 24) had relatively low accuracy in discrimination. Used three pre-processing methods showed similar results except SNV (Group 2 and 4). Can you explain why SNV showed better accuracy than others? Moreover, Conclusions are the proper place where can obtain why and how SNV showed better results than FD or SD. 
Line 438-444 : Better position is Results than Conclusions.
